# Advances in Transversal Topics Applicable to the Care of Bladder Cancer Patients in the Real-World Setting

**DOI:** 10.3390/cancers14163968

**Published:** 2022-08-17

**Authors:** Marga Garrido Siles, Antonio López-Beltran, Paula Pelechano, Ana María García Vicente, Regina Gironés Sarrió, Eva González-Haba Peña, Alfredo Rodríguez Antolín, Almudena Zapatero, José Ángel Arranz, Miguel Ángel Climent

**Affiliations:** 1Oncology Pharmacy Department, Hospital Universitario Virgen de la Victoria, 29010 Málaga, Spain; 2Department of Morphology Sciences, University of Cordoba Medical School, 14004 Cordoba, Spain; 3Radiodiagnosis Service, Fundación Instituto Valenciano de Oncología-IVO, 46009 Valencia, Spain; 4Nuclear Medicine Department, Hospital Universitario de Ciudad Real, 13005 Ciudad Real, Spain; 5Medical Oncology Service, Hospital Universitari i Politècnic la Fe, 46026 Valencia, Spain; 6Pharmacy Service, Hospital Universitario Gregorio Marañón, 28007 Madrid, Spain; 7Urology Service, Hospital Universitario 12 de Octubre, 28041 Madrid, Spain; 8Department of Radiation Oncology, Hospital Universitario de la Princesa, Instituto Investigación Sanitaria IIS-IP, 28006 Madrid, Spain; 9Genitourinary and Gynecologic Section, Department of Medical Oncology, Hospital General Universitario Gregorio Marañón, 28007 Madrid, Spain; 10Medical Oncology Service, Fundación Instituto Valenciano de Oncología-IVO, 46009 Valencia, Spain

**Keywords:** bladder cancer, liquid biopsy, safety drug administration, imaging techniques, immune checkpoints inhibitors

## Abstract

**Simple Summary:**

There are a number of scientific questions that are novel or controversial, which are clinically relevant in the real-world setting for patients diagnosed with bladder cancer, independently of the stage of disease or the histological type and grade of the tumors. These transversal topics have been discussed by a panel of expert specialists who participated in the Genitourinary Alliance Project, aimed to develop actions focused on the integration of relevant information into clinical practice. Advances in radiological imaging techniques have the potential of improving accuracy of staging methods, resulting in a more personalized planning and therapeutic option. The use of liquid biopsy will undoubtedly contribute to an increase in the efficiency of the evaluation of the clinical response and outcome of the disease. It is important to consider specific conditions of elderly people with bladder cancer, as well as the implementation of appropriate measures to enhance safe drug administration.

**Abstract:**

Recommendations regarding transversal topics applicable to bladder cancer patients independent of tumor grade and stage were established by members of the Spanish Oncology Genitourinary Multidisciplinary Working Group (SOGUG). Liquid biopsy in urine and blood samples is useful in the surveillance of non-muscle-invasive and muscle-invasive bladder cancer, respectively. Multiparametric MRI is an accurate, faster and non-invasive staging method overcoming the understaging risk of other procedures. The combination of FDG-PET/MRI could improve diagnostic reliability, but definite criteria for imaging interpretation are still unclear. Hospital oncology pharmacists as members of tumor committees improve the safety of drug use. Additionally, safety recommendations during BCG preparation should be strictly followed. The initial evaluation of patients with bladder cancer should include a multidimensional geriatric assessment. Orthotopic neobladder reconstruction should be offered to motivated patients with full information of self-care requirements. Bladder-sparing protocols, including chemoradiation therapy and immune checkpoints inhibitors (ICIs), should be implemented in centers with well-coordinated multidisciplinary teams and offered to selected patients. The optimal strategy of treatment with ICIs should be defined from the initial diagnostic phase with indications based on scientific evidence. Centralized protocols combined with the experience of professional groups are needed for the integral care of bladder cancer patients.

## 1. Introduction

The increasing availability of novel treatment options and improved oncological outcomes of patients with urothelial bladder cancer have contributed to a recent change towards the challenge for clinical implementation of personalized medicine. Despite impressive advances in tumor biology in terms of genetic mutations and molecular subtypes providing the possibility of tailor-made therapeutic approaches for patients suffering from bladder cancer, a lot of questions remain to be answered. Although inter- and intratumoral heterogeneity represents a difficulty for the clinical implementation of personalized medicine, there are number of transversal topics that are relevant in the real-world setting and are interesting to be considered in the broad overview of bladder cancer patients. Some of these transversal topics in particular have been discussed by a panel of expert specialists, all of whom are members of the Spanish Oncology Genitourinary Multidisciplinary Working Group (SOGUG), and participated in the Genitourinary Alliance Project (12GU), aimed to develop consensus and recommendations focused on the integration of relevant information into the care of bladder cancer patients in daily practice conditions. Therefore, a set of cross-cutting topics applicable to different aspects of the management of patients with bladder cancer were discussed and selected with the aim of improving integrated cancer care, supporting specialists in a fast-changing scientific environment, and integrating scientific advances into the continuous improvement in the care and outcome of bladder cancer.

## 2. Role of Liquid Biopsy

The term liquid biopsy refers to the use of blood and other bodily fluids as a surrogate of tissue samples for diagnosis purposes [1]. Materials for analysis are circulating tumor cells (CTC), exosomes, or circulating free DNA (cfDNA) in blood samples, and shed cells, urinary exosomes, and nucleic acids in urine samples. The main applications of liquid biopsy include early diagnosis, staging, prognosis, follow-up, and assessment of prognostic or predictive biomarkers [1]. On the other hand, the main limitations of using blood samples for liquid biopsy are related to high dilution of diagnostic materials, short half-life in circulation, and challenging isolation of materials, whereas limitations of urine samples are contamination by normal cells, critical sample storage, and contamination by substances affecting PCR reactions [1].

Recent research of diagnostic methods has focused on extracellular vesicles (EVs) that can be found in various bodily fluids. The term EVs includes all types of vesicles that exist in the extracellular space, including microvesicles, exosomes, apoptotic bodies, oncosomes, and prostasomes [2]. EVs are complex biomolecules that carry multiple types of membrane-bound proteins (tetraspanins, Rab family proteins, Tsg101, Alix) and nucleic acids (mRNA, long/short noncoding RNAs, microRNA, DNA) and have emerged as ideal targets for liquid biopsy because they are covered by a lipid bilayer, which can protect them from degradation [2].

Liquid biopsy is a tool for molecular profiling given the high number of non-synonymous somatic mutations reported in bladder cancer [3]. Blood or urine specimens are obtained, which include CTC released by the tumor and cell-free DNA released by apoptotic or necrotic tumor cells, and after processing the tumor is characterized by several molecular technologies, such as whole genome/exome sequencing, targeted next-generation sequencing (NGS), RNA sequencing, immunostaining, multiplex qRT-PCR, or transcriptomics and epigenomics assays [4].

However, clinical implementation of liquid biopsy presents challenges related to the robustness of samples, standardized and optimized processing, and sequencing levels of detection [4] (Figure 1).

The clinical usefulness of liquid biopsies in the clinical management of bladder cancer [4] includes: (a) improvement of clinical staging, especially in radiographically occult regional disease, or detection of tumor burden in patients with advanced disease; (b) improvement of the selection of patients for treatment, such as identification of higher risk patients who would benefit from neoadjuvant chemotherapy or those with probable tumor progression after primary treatment; (c) analysis of mutations as NGS has shown a high level of correlation to tissue mutation burden, detection of therapeutic targets similar to those observed in tissue; and other omics profiling can also be detected by liquid biopsy. However, ongoing clinical trials of CTC and tumor cell (tcDNA) assays as biomarkers in bladder cancer are needed to further demonstrate their utility to improve clinical response and survival. In the review of Kouba et al. [4] of the current status of liquid biopsies in the clinical management of bladder cancer, a summary of clinically relevant studies of CTC and cfDNA with key findings was reported, but in most studies, a limited number of samples were analyzed. However, Christensen et al. [5] used liquid biopsy analysis of FGFR3 and PIK3CA hotspot mutations in a cohort of 363 patients with non-muscle-invasive bladder cancer (NMIBC) and found that high levels of FGFR3- and PIK3CA-mutated DNA in urine and plasma were indicative of later progression and metastasis.

Biomarkers detected in urine by liquid biopsies have shown usefulness in clinical practice for detecting primary and recurrent tumors, they are mainly used for the study of NMIBC (liquid biopsy based on blood samples are preferable in muscle-invasive bladder cancer (MIBC)), DNA/RNA urinary biomarkers account for most molecular alterations found in primary tumors, monitoring recurrence after primary treatment is the main current clinical application, the use of protein-derived urinary biomarkers is limited, and prognosis and identification of new therapeutic targets is an important area of research of DNA/RNA urinary biomarkers. A review of Lopez-Beltran et al. [6] reported useful data of the sensitivity and specificity of tumor-derived DNA and RNA as urine biomarkers for diagnosis and/or surveillance collected in studies published in the literature, as well as data of commercially available urine biomarkers tests based on RNA/DNA alterations. The advantages and disadvantages of sources of biomarkers in liquid biopsies [1] are summarized in Table 1.

Additionally, based on the interactions between circulation exosomal PD-L1 and T cells, it has been reported in melanoma patients that tracking the levels of circulating exosomal PD-L1 may help to predict a patient’s response to anti-PD-L1 therapy (pembrolizumab) [7]. The magnitudes of the increase in circulating exosomal PD-L1 during early stages of treatment, as an indicator of the adaptive response of the tumor cells to T cell reinvigoration, was found to stratify clinical responders from non-responders [8]. Whether these findings may be extensive to patients with urothelial carcinoma is still unknown.


**
*Recommendations and challenges:*
**
Liquid biopsy has great potential in the diagnosis, prognosis and treatment of bladder cancer, but further clinical studies are needed to confirm the efficiency of this technique in the assessment of clinical response and overall survival.Liquid biopsy of urine samples is recommended in the surveillance of NMIBC and liquid biopsy of blood samples in MIBC (e.g., ultrasensitive assays based on real-time PCR capable of detecting trace amounts of the most common genetic and molecular alterations and protein aberrations in NMIBC and MIBC).Cost reduction of commercially available kits is indispensable for the full incorporation of liquid biopsies in routine clinical care of bladder cancer patients.Integration of all available radioimaging, histopathological, liquid biopsy, and molecular medicine data in multidisciplinary committees will change the diagnostic and therapeutic paradigm of bladder cancer.


## 3. Current Perspectives of Radiological Imaging Techniques

Ultrasound (US) is a well-accepted, cost-effective, and noninvasive diagnostic method for screening of patients with suspected bladder cancer, although accuracy depends on several factors, such as distension of the bladder, tumor size, and morphology and location of the lesions. Ultrasound may be inconclusive in the presence of bladder clots, surgical wall changes, and wall trabeculation and has a low sensitivity in the detection of small tumors (<1 cm). Contrast-enhancement ultrasound (CEUS) improves the bladder cancer detection rate of US, especially in US studies that are indeterminate. In a study of 43 patients with suspected bladder cancer, accuracy in the detection of the presence or absence of bladder cancer was higher for CEUS (88.4%) as compared with US (72.1%), but for tumors larger than 5 mm, the sensitivity was very high (94.7%), in contrast to a very low sensitivity (20%) for tumors smaller than 5 mm [9]. Multidetector computed tomography urography (MDCTU) has the highest diagnostic accuracy of the available imaging techniques for diagnosing upper urinary tract urothelial tumors, with a sensitivity of 0.97–1.0 and a specificity of 0.93–0.99 [10]. Magnetic resonance (MR) urography is indicated in patients who cannot undergo MDCTU when radiation or iodinated contrast media are contraindicated.

In relation to tumor staging of bladder cancer, MR urography is increasingly being used for the high spatial resolution and soft-tissue contrast to delineate tumors from the normal detrusor muscle, making MRI superior to computed tomography (CT) for differentiating non-muscle-invasive from muscle-invasive bladder tumors. Muscle invasion is usually better delineated with T2-weighted MRI than with CT, and the muscular (detrusor) layer appears as a hypointense band against hyperintense intraluminal urine and perivesical fat [11]. The use of functional MRI techniques, such as diffusion-weighted imaging (DWI) and dynamic contrast-enhanced MRI (DCE-MRI) helps to increase the accuracy of the MRI. DWI provides functional information on the cellularity of tumors based on the gradient of free diffusion of water molecules into the intracellular and intercellular spaces, whereas DCE-MRI enables alterations of tumor vascularization to be assessed [12]. Moreover, important improvements in MRI technology have led to the introduction of multiparametric MRI (mpMRI), combining anatomic and functional sequences. mpMRI is currently the most accurate imaging modality for local staging of bladder cancer, capable to accurately distinguish MIBC from NMIBC [13]. In 2018, a panel of expert multidisciplinary team members developed “Visual Imaging Reporting Data System” (VI-RADS), a standardized approach to imaging and reporting mpMRI for bladder cancer. VI-RADS is an MRI-based scoring system of bladder tumors for predicting MIBC using T2, DWI, and DCE-MRI weighted images [14]. A five-point scale is used for scoring each method, which are then combined to derive an overall VI-RADS score. The likelihood of MBIC is classified into five categories (from VI-RADS score 1–2 which means that MIBC is very unlikely to score 4–5 which means that MIBC is likely) [14]. A meta-analysis of six studies evaluated the value of VI-RADS for predicting MIBC and showed a pooled sensitivity and specificity of 0.83 and 0.90, respectively [15].

In reference to nodal staging of bladder cancer, conventional CT and MRI showed critical limitations for identifying lymph node metastases based on size criteria only, both techniques are unable to identify metastases in normal nodes and to distinguish large hyperplastic (benign) nodes from malignant nodes [16]. Functional MRI enhanced with ultrasmall superparamagnetic iron oxide contrast agent (USPIO) is a promising imaging modality in the detection of lymph node metastases [16].

Bone scintigraphy is indicated in patients with specific symptoms or signs suggestive of bone metastases, although MRI (DWI) is more sensitive and specific for diagnosing bone metastases than bone scintigraphy [17]. Current guidelines [18], however, have not incorporated MRI because of limitations related to availability and cost-effectiveness of the technique. However, MRI may be especially useful in the characterization of lesions that are considered indeterminate on CT or bone scintigraphy, or in the assessment of early bone marrow infiltration. CT and MRI are the diagnostic techniques of choice to detect lung and liver metastases, respectively [17]. In clinical practice, CT is the imaging technique used for staging and follow-up of MIBC.

Additionally, CT is the standard technique in the assessment of response to treatment. The Response Evaluation Criteria in Solid Tumors (RECIST) [19], published in 2009 and based on reduction of tumor size and absence of new lesions, has been a widely accepted method for assessing tumor response. In 2017, the RECIST working group developed immune RECIST (iRECIST) [20] for use in cancer immunotherapy trials, with definitions of hyperprogression, pseudoprogression, and progressive disease.

Radiomics is a new field of quantitative image analysis based on the concept that biomedical images may reflect underlying pathophysiological relationships, and in which digital medical images, such as MRI, PET, or CT examinations, are converted into mineable high-dimensional data [21]. Diagnosis, prognosis, and predictive accuracy may be improved according to models developed by combining radiomics and other patient data using refined bioinformatics tools. Radiomics showed an exponential development in recent years, in particular in oncological imaging. In bladder cancer, radiomics has shown promising results in the assessment of the grade of tumor aggressiveness and local staging [22] (Figure 2).


**
*Recommendations and challenges:*
**
mpMRI is more accurate, faster, and a non-invasive staging method, overcoming the risk of understaging associated with transurethral resection of bladder tumors (TURBT) (such as differentiation of NMIBC from MIBC, diagnosing T3-T4 disease).MRI preoperatively could help in planning treatment, including radical complete TURBT, identification of patients without need of re-TURBT, patients suitable for bladder-sparing and chemoradiation, and replacing TURBT with mpMRI in selected patients (poor fitness).The need of frequent repeat cystoscopies at follow-up has driven mpMRI use as a non-invasive alternative.Standardization of MRI acquisition and validation in prospective studies is needed to ensure MRI reproducibility.Imaging-based biomarkers of treatment response should be explored in further studies.


## 4. Future Perspectives of Nuclear Medicine Imaging Techniques

In relation to biological parameters, it has been shown that the presence of CTCs (using the CellSearch CTC test) was associated with increased risk of radiological metastatic disease on 18F-fluorodeoxyglucose-positron emission tomography/computed tomography (FDG-PET/CT) in patients with advanced bladder cancer treated with radical cystectomy, and also with earlier progression but not of cancer-specific death [23]. Further studies are needed to define the role of CTC positivity as a method to select patients for neoadjuvant chemotherapy or treatment intensification following radical cystectomy. In a retrospective analysis of 63 patients in which PD-1/PD-L1 status of bladder cancer was predicted based on FDG-PET/CT imaging, patients with both PD-1 and PD-L1 positivity showed significantly higher maximum standardized uptake values (SUVmax) as compared with patients with negative PD-1 and PD-L1 [24]. Accordingly, PD-1/PD-L1 status of urothelial cancer may be predicted by FDG-PET/CT. Additionally, FDG-PET/CT may be useful to assess the most appropriate therapeutic approach.

Moreover, simultaneous PET/MRI has been suggested to have a possible role to help clinicians in the management of bladder cancer. In a prospective pilot study, 22 patients underwent MR urography and FDG-PET/MRI for the evaluation of bladder cancer, and the information obtained by simultaneous PET helped to refine suspicion of tumor in all intrapelvic anatomic sites evaluated, particularly at the level of the pelvic lymph nodes, which could be associated with an improvement of clinical staging [25]. 

Other metabolic radiotracers used for PET/CT include C-11 labeled tracers, such as 11C-choline, 11C-acetate, and 11C-methionine. In the case of 11C-choline and 11C-acetate, the main advantage is their low urinary excretion, although the performance of 11C-methionine is not superior to other radiotracers.18F-NaF is used only for the detection of bone disease with a diagnostic accuracy higher than standard bone scintigraphy with 99mTc-based radiotracers. The prostate-specific membrane antigen (PSMA) is expressed in the neovasculature of a number of solid malignancies including urothelial carcinoma. PSMA-targeted agents such as 18F-DCFPyL may offer improved image quality relative to other conventional modalities but the relatively scant expression of PSMA by urothelial cancer limits the utility of 18F-DCFPyL imaging of this malignancy [26].

Imaging techniques are unable to detect microscopic lymph node disease, and about one-third of patients with MIBC have lymph node infiltration (between 30% in T2 and 50% in T3 tumors). However, lymph node drainage is variable and unpredictable, with crossover lymphatic drainage in 45% of cases, and there is controversy regarding the extent of pelvic lymphadenectomy. In a systematic review and meta-analysis of sentinel node biopsy in bladder cancer based on eight studies and 336 patients with MIBC at clinical stage cT1-T4N0M0, the pooled detection rate was 91% and the pooled sensitivity 79%; however, by omitting studies that enrolled >50% of patients at a pT stage of 3 or 4, the pooled sensitivity increased to 93%. Based on the existing studies, pT1 or pT2 bladder cancer patients with clinically negative lymph nodes are the most appropriate group for sentinel lymph node mapping [27].


**
*Recommendations and challenges:*
**
The role of FDG-PET/CT for predicting tumor biology and prognosis is still uncertain, although results of a relationship between metabolic imaging and CTC positivity and PD-1/PD-L1 status are promising.The combination FDG-PET/MRI could increase diagnostic reliability and improve staging of patients, but methodological aspects and criteria for the interpretation of images have not been definitively established.The sentinel lymph node biopsy approach with radiocolloids in urothelial carcinoma is justified but the clinical value of the procedure is unclear, with a higher diagnostic accuracy in pT1-T2 tumors.


## 5. Safety Considerations in Oncological Patients

From a safety perspective, cancer patients are vulnerable and complex to manage because of a series of interrelated factors (Table 2).

In this context and from a wide perspective of safety, main actions taken by the oncology pharmacy service for improving safety and optimizing pharmacotherapy include detection and prevention of drug interactions, promoting medication reconciliation programs, improvement of patients’ adherence, safe drug manipulation, and validation of drug prescription. The main predisposing factors to drug interactions are the high number and many different kinds of drugs used to treat cancer, the complex pharmacological profile of antineoplastic drugs, the deterioration of physical function of cancer patients, and the frequent use of phytotherapy.

Drug–drug interactions (DDIs) comprise an important problem in medical oncology practice, and a systematic review of eight studies showed a large frequency of DDIs ranging between 12% and 63% [28]. In a computer-based medication prescription system for dispensing oral anticancer drugs to 898 outpatients in three Dutch centers, DDIs were identified in 46% of patients with coumarins and opioids as the drug classes most frequently involved [29]. A large proportion of the identified DDIs in cancer patients are of minor clinical significance, but some potential drug interactions may seriously affect efficacy/safety of treatment [29,30]. Additionally, 2% of unplanned admissions in cancer patients are considered to be associated with a DDI [31,32]. In some cases, DDIs may be masked by other symptoms or adverse drug reactions.

Cancer patients are also at risk of food–drug interactions, and serious patient education pitfalls in understanding treatment compliance have been identified [33]. On the other hand, use of complementary and alternative medicine (CAM) has increased in the last decade in both palliative and curative settings, with different reasons argued by patients for the use of these compounds [34,35]. Although most patients were not exposed to any significant risk of harm from interactions with conventional medicines, the efficacy of treatment or an increase in adverse events may be seen in some patients due to the combined use with herbal products [36].

Main actions to increase safety in relation to DDI are the following: systemic review of all medications taken by the patient, analysis of interactions and assessment of the clinical significance, and definition and efficient communication of recommendations for patients and healthcare professionals.

The increased use of oral cancer therapy (more than 25% of treatments in oncological patients) has been a great advance through improved outcomes and convenience of treatment administration, although long-term medication adherence is one of the challenges with this treatment modality. In a sample of 181 patients prescribed oral targeted therapy or chemotherapy, participants took only 85% of the oral therapy on average, with different percentages of poor adherence by cancer type (23.3% for hematological cancer and 42.9% for sarcoma) [37]. In a study of patients treated with oral chemotherapy, in which patient-reported outcomes (PROs) were considered, adherence of less than 80% was shown in 30% of patients, with adverse effects or concerns about adverse effects reported as the most frequent reason for non-adherence [38]. In this respect, it is important to consider the patient’s adherence to all prescribed medications for the influence of the correct use of the anti-neoplastic regimen on the outcome, as well as the increasing incorporation of oral targeted therapies as a result of advances in molecular biology. Early discontinuation of medication and non-adherence has been associated with poor outcomes and higher use of healthcare resources [39,40]. Different strategies focused on factors related to the patient, the treatment, and the healthcare system (Figure 3) should be considered to improve adherence rates and control of the disease. Intensified pharmaceutical care in the framework of multidisciplinary care can enhance adherence and prolong treatment to oral antineoplastic regimens [41]. During the entire treatment process, it is important to ensure easy communication between patients and their healthcare professionals in order to solve doubts and to assess and reinforce adherence to treatment.

Drug-related iatrogenesis can be reduced by medical reconciliation, which facilitates the transmission of exhaustive data at care transition points. Medical reconciliation can be associated with important clinical benefits given the involvement of different specialists and levels of care in the treatment of patients with bladder cancer and the vulnerability of these patients to adverse drug effects. Data of 14 studies included in a systematic review showed that medical reconciliation led to the identification of drug-related problems in 94.7% of patients and discrepancies in 88% [42]. The advantages of medical reconciliation strategies include identification of discrepancies and reduction in the incidence of medication errors, improvement of adherence, reduction of medication-associated adverse events, and positive impact in the reduction of readmissions after discharge and post-hospital healthcare utilization [43,44,45].

Hospital oncology pharmacists play a central role in the validation of prescription of anticancer treatments, checking a series of aspects to ensure that the prescribed treatment is appropriate, including treatment days, duration, and interval between cycles according to the protocol; assessment of all drugs at appropriate doses and adjusted to laboratory data and previous toxicity; assessment of supportive treatment and premedication to prevent and reduce adverse events; pharmaceutical formulations adequate to the patient’s condition (monitor swallowing problems, recommendations on safe handling); review and prevention of interactions; encouragement of adherence to treatments, and reinforcement of information [46].


**
*Recommendations and challenges:*
**
An adequate prescription validation process that finally results in the administration of the prescribed treatment targeted to the established protocol and the patient’s clinical characteristic and individual laboratory data is an important step that contributes to safety of the oncological patient.Based on the large volume of cancer patients, frequency of polypharmacy, care of the patient in different healthcare levels, medication reconciliation protocols according to the profile and treatment of patients should be established.Involvement of a multidisciplinary team and improvement of communication of potential drug–drug interactions is important to ensure safety management of cancer patients.The presence of hospital oncology pharmacists as members of the tumor committees would contribute to improve safety of the drug use in cancer patients.


## 6. Safety in the Administration of Treatment

The term “hazardous drug” was introduced in 1990 by the American Society of Hospital Pharmacy and used for the first time in an alert of the National Institute for Occupational Safety and Health (NIOSH) in 2004, referring to drugs that are known to cause harm in healthcare workers due to their inherent toxicity. NIOSH defines a hazardous drug as a drug that exhibits one or more of the following toxicity characteristics: carcinogenicity, teratogenicity, reproductive toxicity, organ toxicity at low doses, genotoxicity, and new drugs with structure and toxicity profiles to drugs defined as hazardous according to the aforementioned criteria.

It is important to distinguish two types of risk: (a) chemical risk associated with chemical properties of most hazardous drugs, and (b) biological risk, which is a potentially infectious risk (e.g., bacillus Calmette-Guérin (BCG) for intravesical immunotherapy for treating early-stage bladder cancer). Exposure to hazardous drugs may occur during any of the different processes of reception, storage, manufacturing, transportation, administration, and waste disposal, although manufacturing and administration to a lesser extent are associated with the highest exposure risks. Inhalation and direct contact of skin with contaminated surfaces are the main exposure routes, followed by the oral route (hand-to-mouth transmission), and parenteral route (puncture, cut, sharp material, needles).

The principal sources available for information regarding hazardous drugs and their classifications are shown in Figure 4.

On the other hand, factors associated with the risk of exposure to hazardous drugs include the intrinsic risk of the drug, the pharmaceutical formulation and the route of drug administration (the intravesical route is one of the most risky), the activity being performed (manufacturing involves a higher risk, although the level of protection is also higher), and the frequency of exposure (Figure 5).

In relation to BCG, recommendations to reduce the risk of exposure include centralized preparation in the service of pharmacy inside a biological safety cabinet, avoidance of cross-contamination due to the high risk of a contaminated solution in immunosuppressed patients, coordination of BCG preparation with the service of urology due to the limited stability of liquid BCG, BCG preparation with closed systems and without the use of needles, and a closed system administration set should be used. Likewise, chemotherapeutic agents and immunotherapy drugs are considered hazardous drugs. Biological safety cabinets should be used for preparation, and closed systems for administration. Exposure of healthcare workers during drug administration can be minimized using closed systems and the personal protection equipment individualized to the risk (Table 3).


**
*Recommendations and challenges:*
**
To improve safety in the use of drug treatment in patients with bladder cancer, in particular intravesical BCG therapy, diffusion of information and especially training of healthcare personnel are essential factors.It is important to implement appropriate measures during drug administration to prevent (or minimize) exposure of the healthcare personnel.BCG is a hazardous drug with biological risk and involves a potential risk in immunosuppressed patients, so safety recommendations during preparation should be carefully followed.


## 7. Geriatric and Vulnerability Assessment of the Patient with Bladder Cancer

Cancer can be considered an age-related disease because the incidence of most cancers increases with age, with persons over 65 accounting for 60% of newly diagnosed malignancies (30–40% in patients older than 70 years), and cancer-related deaths are expected to occur in the elderly population. Ageing is a heterogeneous process and geriatric deficits associated with a reduction in life expectancy and higher toxicity of anticancer treatments cannot be detected by standard oncological assessment. Some characteristics of bladder cancer in elderly individuals are shown in Table 4.

The provision of geriatric assessment (GA) with management recommendations to oncologists can reduce toxicity in older patients with advanced cancer treated with chemotherapy and/or other agents. In a cluster randomized clinical trial of 718 patients (mean age 77 years), the use of GA information reduced the proportion of patients with grade 3–5 toxicity from high-risk palliative cancer treatment (50% vs. 71% in the usual care group) without compromising overall survival [47]. In a randomized controlled trial of 600 patients ≥65 years diagnosed with a solid malignancy and starting a new chemotherapy regimen, the incidence of grade 3–5 chemotherapy-related toxicity was 50.5% in the geriatric assessment-driven intervention and 60.4% in the standard of care arm, without differences in emergency room visits, hospitalizations, and average length of stay [48].

A recent position paper of the International Society of Geriatric Oncology (SIOG) emphasizes that age should not determine treatment, patients should be fully involved in decisions and undergo cognitive impairment evaluation and comprehensive geriatric assessment, and standard treatment for fit patients should be provided [49]. Elderly patients should be managed according to their individual health status and not according to age, and geriatric assessment using geriatric screening tools (G8, Mini-COG™, the Vulnerable Elders Survey-13, geriatric depression scale, activities of daily living, etc.) should be implemented in daily practice as a previous evaluation of candidates to active treatment (surgery, neoadjuvant/adjuvant chemotherapy, treatment of metastatic disease) [50].

The importance of geriatric evaluation in elderly cancer patients has been emphasized because findings may reveal impairments that have passed undetected in the medical history and/or physical examination [51]. Additionally, results of geriatric assessment may influence treatment decisions or predict potential toxicity of oncological treatment and outcomes [51].

Functioning, comorbidity, mental and cognitive status, fatigue, nutrition, geriatric syndromes, social relationships, and social support are some of the domains that should be included in the geriatric assessment. Different instruments can be used for the evaluation of these domains, the choice of which depends on preferences of the medical team, characteristics of the instruments, and availability of questionnaires [51]. Hurria and colleagues [52] have proposed a computer-based geriatric assessment instrument that has been shown to be feasible and valid to capture data in elderly patients with cancer and was also preferred to traditional paper-and-pencil evaluation. 

Although the integration of geriatric assessment into oncology clinics is still limited, a simple geriatric assessment can be performed in any clinical setting, regardless of the available resources, and will provide valuable data of the physiological age of older people with cancer [53].


**
*Recommendations and challenges:*
**
Chronological age alone is not a reliable indicator of the functional and physiological status of people over 65 years of age with cancer and should not be a main factor for guiding treatment decisions in patients with bladder cancer.Elderly patients should be referred to the service of geriatrics for multidimensional evaluation of cognition, nutrition, physical function, comorbid conditions, social support, and psychological status.If a geriatrics service is not available, consultation with other services are recommendable to assess the nutrition status, the risk of falls, to establish medication reconciliation, or interventions for patients with deficient social support.Geriatric assessments should be implemented in the initial evaluation of patients with bladder cancer and may constitute a validation system for clinical decision making.


## 8. Bladder Substitution Techniques

Replacement of the urinary bladder aimed at storage of urine at low pressure, preserving renal function, and to achieve adequate voiding and urinary continence is a major challenge in urological surgery. Types of urinary diversion surgery include orthotopic neobladder substitution (Studer “U” or Hautmann “W”), Indiana pouch heterotopic continent diversion, neobladder formed from a segment of ileum (ileal conduit also known as Bricker conduit), cutaneous urinary diversion, and sigma-rectal pouch. Although an orthotopic neobladder reconstruction is an effective treatment option and remains the gold standard at many centers of expertise, the Bricker procedure (ileal conduit urinary diversion) is still performed in 20–25% of patients after radical cystectomy. Age and comorbidity are critical factors in the selection of urinary diversion method, with frail older people more likely to undergo the Bricker ileal conduit diversion. Advantages of ileal conduit urinary diversion include shorter operative time, quicker recovery and ease of care by others, whereas some disadvantages are requirement of external appliance, impairment of body image, parastomal hernia, and peristomal skin irritation.

Motivated patients with realistic expectations, with skills for self-catheterization, a serum creatinine level < 2 mg/dL or glomerular filtration rate > 35/mL/min, and without liver dysfunction (risk of hyperammonemia) are candidates for bladder substitution procedures. Age is not a contraindication, whereas bladder diversion surgery is contraindicated in the presence of urethral tumor invasion, tumor detection in the intraoperative biopsy, and N2–N3 stage. Relative contraindications include inability for self-catheterization, mental incapacity for self-care, complex urethral stenosis, NMIBC in the urethra or bladder neck, previous pelvic radiotherapy or incontinence, and low life expectancy.

In relation to health-related quality of life (HRQOL), most research has focused on comparing HRQOL outcomes between different urinary diversion types, such as ileal conduit vs. neobladder vs. continent cutaneous diversion, the results reported have been mixed and clear evidence-based findings differentiating these techniques have not been found [54].

Today, about 75% of patients treated with radical cystectomy are candidates for orthotopic urinary diversion, and it is reasonable to consider that when no absolute contraindications are present, a small percentage of patients eligible for a continent urinary diversion chose an ileal conduit for personal reasons [55]. However, in clinical practice, less than 20% of patients undergo orthotopic neobladder reconstruction after radical cystectomy. In relation to complications, uretero-ileal stenosis, kidney stones, infections, and metabolic complications can occur in all types of urinary diversion procedures. Parastomal hernia and appliance/skin problems are associated with the ileal conduit, and incontinence (mainly nighttime incontinence), urinary retention, bladder stones, and afferent limb stenosis with orthotopic neobladder. Other complications of orthotopic neobladder reconstruction include ureteral strictures in 4% of cases, hyperchloremic metabolic acidosis, vitamin B12 deficiency, and a 21% increase in the risk of fractures [56]. A study of continence outcomes following radical cystectomy and orthotopic neobladder in 180 male patients who completed a pad usage questionnaire showed that there was a significant improvement in continence by 6 months, and 88% of patients achieved daytime continence by 12 months; the corresponding percentage for nighttime continence was 44%, and in the 18–36 month range, 53% of patients reported both day and nighttime continence [57].

According to recommendations of the European Association of Urology Muscle-invasive and Metastatic Bladder Cancer Guideline Panel, patients should be referred to hospitals in which a high number of radical cystectomies are performed on an annual basis (minimum 10, preferably more than 20), since hospital volume influences postoperative survival and the quality of care provided to bladder cancer patients [58]. In addition, robot-assisted radical cystectomy (RARC) as compared with open radical cystectomy (ORC) is associated with shorter hospitalization and reduced blood loss, although direct costs are greater and operative time longer. Additionally, it has been shown that RARC is non-inferior to ORC for 2-year progression-free survival [59].


**
*Recommendations and challenges:*
**
The choice of orthotopic neobladder reconstructions should be offered to motivated patients, with full information of self-care and self-catheterization needs.It is necessary to discuss the pros and cons of all bladder substitution techniques, even in case that the operation should be performed in another center.It is necessary to implement a centralization plan involving hospitals and surgeons, both for orthotopic neobladder reconstruction and radical cystectomy in centers with annual volumes >20 procedures.Daily practice challenges include implementation of enhanced recovery after surgery (ERAS) protocols, geriatric assessment, appropriate evaluation of HRQOL, and protocolized follow-up after radical cystectomy.


## 9. New Radiation Techniques

Conservative management of infiltrating bladder cancer based on a trimodality approach (TURBT, external radiotherapy and systemic chemotherapy) and the multidisciplinary involvement of different specialties is becoming a real alternative to radical surgery for bladder preservation in selected patients. It has been shown that the best results of the trimodality therapy are obtained using a strategy in which radiochemotherapy follows TURBT. Bladder preservation with radiochemotherapy is supported by numerous guidelines [60,61,62]. No randomized clinical trials comparing radical cystectomy and trimodality treatment have been published, and data from retrospective analyses and phase II clinical studies have been reported. In a critical review of the literature of the ability of trimodality treatment to achieve good disease control in selected MIBC patients, complete response rates ranged around 50–78%, 5-year overall survival around 56%, 10-year disease-specific survival around 65%, and survival with bladder preservation around 50–65% [63]. Long-term follow-up studies have shown that bladder-sparing treatment is a successful approach for muscle-invasive bladder cancer, with maximal TURBT of the bladder tumor and response to induction therapy as the most relevant predictive factors [64].

In relation to HRQOL, bladder-sparing trimodality therapy and radical cystectomy are both associated with good long-term outcomes in MIBC survivors [65], although advantages of trimodality therapy include preservation of sexual function and body image, better social and emotional functioning as well as bowel function, and a gain of quality-adjusted life years (QALYs) relative to radical cystectomy [66].

Prediction of chemoradiation response would enable selection of optimal candidates for bladder-sparing treatment and may ultimately improve prognosis and HRQOL of MIBC patients. The possible role of different biomarkers that can be used for predicting and improving the chemoradiation response and prognosis on bladder preservation therapy has been reported [67], but none of these biomarkers have been validated (Figure 6).

In a study that evaluated the association between MRE11 expression (a protein involved in the DNA double-strand break repair mechanism) and outcome in patients from six NRG/RTOG bladder-sparing protocols, with tissues available from 135 patients and slides analyzed via Automated Quantitative Image analysis (AQUA), it has been shown that low expression of this biomarker was associated with significantly higher disease-specific mortality [68].

A clinical approach of great interest has been to combine radiotherapy with radiosensitizers including synchronous chemotherapy (cisplatin, paclitaxel, 5-fluorouracil, mitomycin C, gemcitabine) [69,70] or hypoxic modifiers (carbogen and nicotinamide) [71], and more recently, radiotherapy with immune checkpoints inhibitors (ICIs) [72]. The role of chemotherapy in the adjuvant [73] or neoadjuvant [74] setting is still unclear. Radiosensitization with immunotherapy seems to stimulate both regional and distal/abscopal immune effects, and preclinical and clinical data from different tumors (localized and metastatic prostate cancer, melanoma, glioblastoma, breast, and lung cancer) and ICIs (durvalumab, ipilimumab, nivolumab, pembrolizumab) support the beneficial effects of the combination of immune-modulation and radiotherapy [75]. Data in bladder cancer patients are lacking, and the best sequencing of neoadjuvant vs. concurrent immunotherapy with ICIs and radiation therapy is still undetermined.

Regarding the integration of next-generation imaging (NGI) and chemoradiotherapy, the best schemes for treatment plans remain unclear, including the role of elective nodal irradiation/treatment volume irradiation, feasibility and efficiency of an hypofractionated approach, the new modality of functional biological imaging to define volumes and contribute to personalized treatment, and adaptative radiotherapy (ART) in dose escalation focal radiotherapy.


**
*Recommendations and challenges*
**


Bladder-sparing protocols based on trimodality therapy should be implemented in centers with well-coordinated multidisciplinary teams and should be offered to selected patients with an appropriate clinical profile.The use of biomarkers combined with imaging technologies (imaging markers and functional MRI) is a promising approach in the management of bladder cancer patients.Optimal schemes of immunotherapy as sensitizers for radiotherapy are still undefined but moderate hypofractionated radiation therapy seems to be a beneficial approach. Extreme hypofractionation/immunotherapy is not advisable due to the high toxicity.Based on a potential higher sensitivity to some ICI drugs, the use of small radiation therapy volumes is recommended.

## 10. Integration of Immunotherapy in the Multidisciplinary Management of Bladder Cancer

Immunotherapy with PD-1/PD-L1 agents is changing the landscape of treatment in the management of solid tumors including urothelial carcinoma. General classification of patients with bladder cancer and treatment approach for NMIBC, MIBC, and metastatic disease are shown in Figure 7.

Immunotherapy is integrated in the management of patients with bladder cancer and there are ongoing clinical trials in NMIBC after TURBT, in the perioperative setting of MIBC, in first-line treatment of metastatic disease in patients eligible for cisplatin chemotherapy, and unfit patients treated with carboplatin chemotherapy. In patients with metastatic disease treated with cisplatin-based chemotherapy, the use of immunotherapy has shown to increase disease-free survival. Immunotherapy as a consolidation of maintenance treatment after first-line chemotherapy and in second-line treatment of metastatic disease has shown to increase overall survival (e.g., pembrolizumab, nivolumab, avelumab, and atezolizumab). However, although PD-1/PDL-1 drugs have shown the potential therapeutic benefits, toxicity and high financial costs should be considered together with the fact that robust evidence from randomized clinical trials is still insufficient [76], although nivolumab has received FDA approval for the adjuvant treatment of patients with urothelial carcinoma at a high risk of recurrence following radical resection regardless of prior treatment with neoadjuvant chemotherapy, nodal involvement, or PD-L1 status [77].

Finally, circulating tumor DNA (ctDNA) testing is a promising minimally invasive technique to identify patients at risk of recurrence after surgery of bladder cancer receiving adjuvant immunotherapy. In the framework of the phase III IMvigor010 trial, patients in the atezolizumab arm who were positive for ctDNA had improved disease-free survival and overall survival versus the observation arm, whereas no difference in disease-free survival or overall survival between treatment arms was noted for patients who were negative for ctDNA [78].


**
*Recommendations and challenges:*
**
Indications of immunotherapy for the management of metastatic bladder cancer should be based on evidence rather than on availability of drugs.The optimal strategy of treatment with ICIs should be defined from the initial diagnostic phase, with clear identification of clinical entities of bladder cancer and their corresponding diagnostic and therapeutic circuits (essential and optional clinical and histopathological tests for the use of ICIs, treatment approaches and sequence of healthcare professionals involved in the patient’s care).Definition of alternatives when the better option is not available is also necessary.Patients should be fully informed regarding access and difficulties to different approved ICI alternatives.


## 11. Conclusions

The advances in the research of radiological imaging techniques in patients with bladder cancer have made it possible to have more precise and less invasive staging methods, which could lead to better treatment planning with more personalized therapeutic options in selected patients. High-resolution micro-ultrasound (mUS) technology has been introduced very recently and the initial results indicate that mUS may have a better performance than MRI in distinguishing NMIBC from MIBC [79]. Recently, new hybrid tracers such as indocyanine green (ICG)-99mTc-nanocolloid have been developed, which have also been applied to bladder cancer [80], although these are very preliminary studies. The role of nuclear medicine imaging techniques in the prognosis, diagnosis, and staging of patients has great potential, but to date there are still aspects that need to be clarified. Similarly, a better understanding of liquid biopsies has provided the background for its incorporation into clinical practice awaiting results that confirm its efficiency in the evaluation of clinical response and survival.

On the other hand, the improvement of safety and optimization of pharmacological treatments requires an adequate review and reconciliation of medication, management and prevention of interactions and a multidimensional, systematic, and protocolized evaluation in older patients, which facilitate appropriate adherence to the therapeutic plan, optimizing its efficacy and reducing its toxicity. The use of ICIs in the treatment of patients with bladder cancer should be based primarily on the available evidence for the indications and effectiveness of these drugs at each stage of bladder cancer.

In relation to safety, it is also necessary to consider the implementation of appropriate measures that would allow a safe drug administration both for the patients and the healthcare personnel involved in drug treatment.

Finally, patients should be appropriately and extensively informed of the currently available new surgical and radiotherapeutic options. The SOGUG group recommends the need to establish well-coordinated multidisciplinary teams and, in some cases, the implementation of centralized protocols in some hospitals according to characteristics of the center and experience of the professional groups.

## Figures and Tables

**Figure 1 cancers-14-03968-f001:**
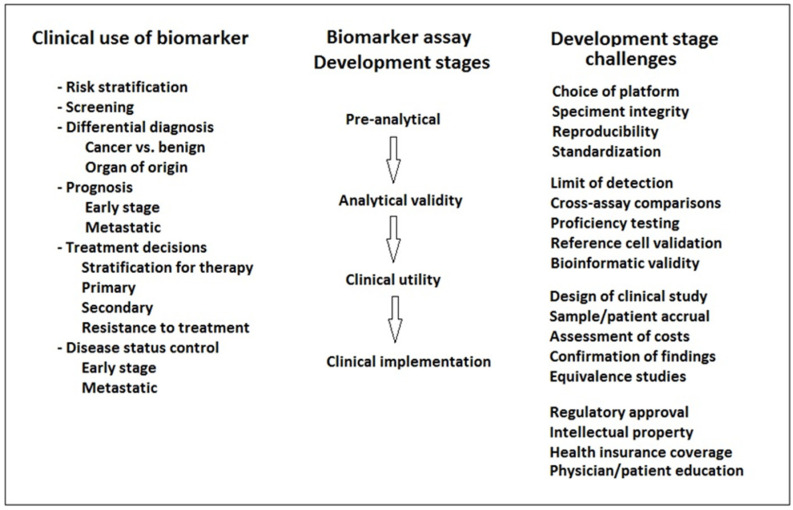
Steps in the implementation of liquid biopsy biomarkers for the detection of bladder cancer.

**Figure 2 cancers-14-03968-f002:**
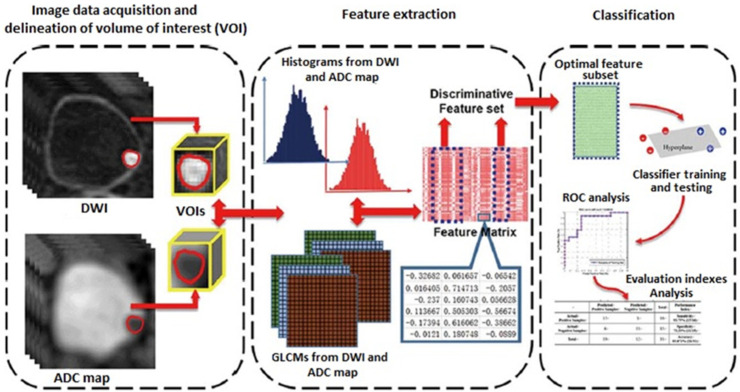
Radiomics strategy for bladder cancer staging (DWI: diffusion-weighted imaging; ADC: apparent diffusion coefficient; GLCM: gray-level co-occurrence matrix; ROC: receiver operating characteristics curve.

**Figure 3 cancers-14-03968-f003:**
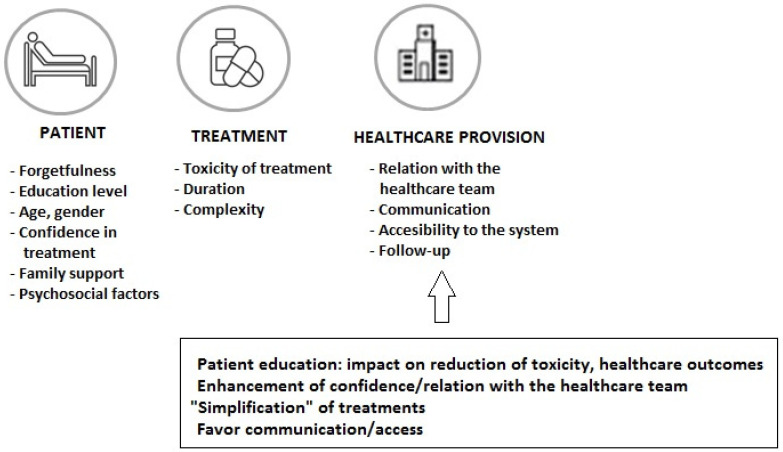
Factors related to the patient, the cancer treatment, and the healthcare system involved in non-adherence to oral anticancer medications.

**Figure 4 cancers-14-03968-f004:**
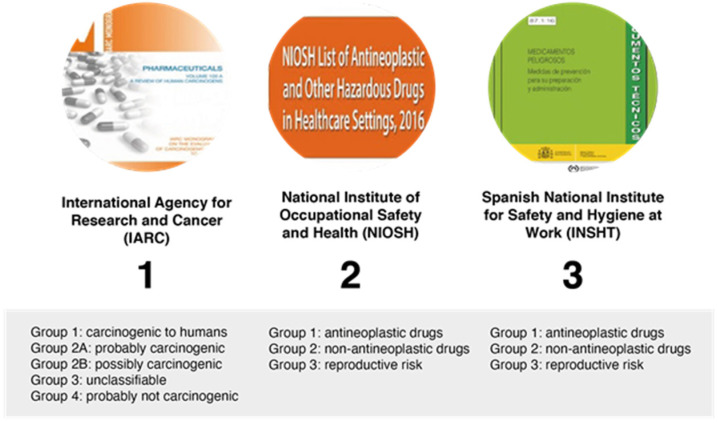
Classification of hazardous drugs by different organizations.

**Figure 5 cancers-14-03968-f005:**
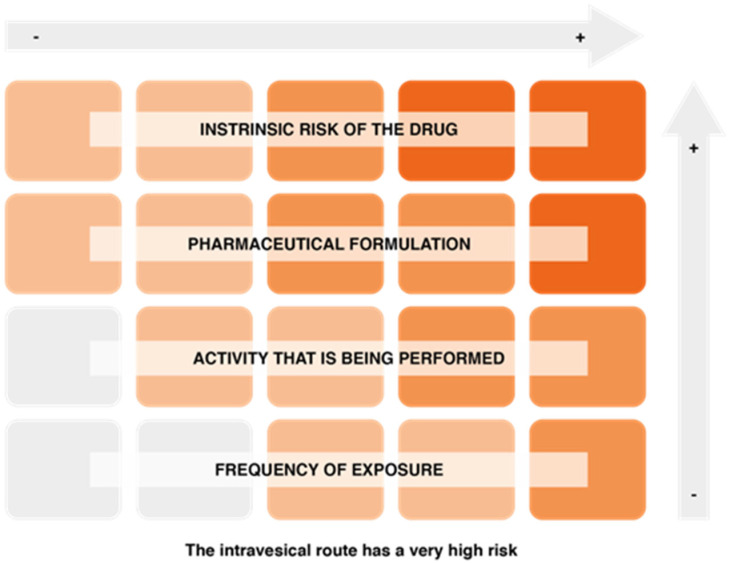
Factors conditioning the risk of exposure to hazardous drugs.

**Figure 6 cancers-14-03968-f006:**
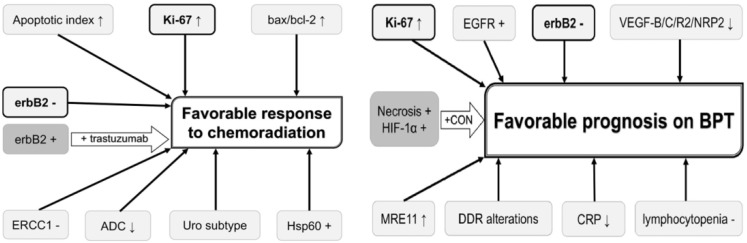
Potential useful biomarkers for improving prognosis in bladder preservation strategies (BPT: bladder preservation therapy).

**Figure 7 cancers-14-03968-f007:**
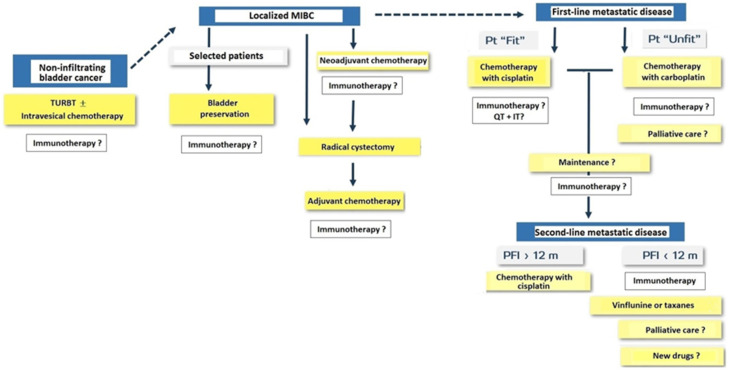
Treatment approach in patients with non-infiltrating bladder cancer, muscle-invasive bladder cancer (MIBC) and metastatic disease (Pt: patient; TURBT: transurethral resection of bladder tumor; QT: chemotherapy; IT: immunotherapy; PFI: progression-free interval; m: months).

**Table 1 cancers-14-03968-t001:** Characteristics of sources of biomarkers in liquid biopsies [1].

Source	Advantages and Disadvantages
Blood cell-free DNA	− PROs: easy tool for tumor DNA characterization; highly sensitive commercially available kits based on dPCR − CONs: high dilution of tumor-derived DNA; low specificity of dPCR
Blood circulating tumor cells (CTCs)	− PROs: actual identification and calculation of whole tumor cells − CONs: low number of circulating CTCs; platelet masking; challenging isolation and capturing
Blood exosomes	− PROs: identification of proteins, RNA and microRNA − CONs: challenging exosome isolation; few commercial kits available
Urinary DNA	− PROs: higher concentration of tumor-derived DNA from urological and non-urological malignancies. Several commercial kits available for methylation and mutation profiles − CONs: fragmented DNA (50–100 bp); critical sample storage (need of prompt refrigeration)
Urinary cells	− PROs: easy isolation from urinary sediment − CONs: contamination by normal cells, critical sample storage
Urinary cells	− PROs: identification of proteins, RNA, microRNA − CONs: difficult isolation of exosomes from urine, unavailability of commercial kits for in vitro diagnostics

**Table 2 cancers-14-03968-t002:** Factors associated with safety concerns in cancer patients.

− Antineoplastic drugs are classified as hazardous drugs due to the high rate of morbi-mortality related to medication errors
− Inherent toxicity of antineoplastic treatments
− Treatment schemes are complex, with constant incorporation of new drugs
− Cancer patients are progressively older with underlying comorbidities and are polymedicated
− Cancer patients have diminished physiological reserves
− Different specialists and healthcare levels are involved in the care of cancer patients, who also receive multimodal therapies (surgery, chemotherapy, radiotherapy, etc.)
− An increasing number of drugs are administered, with an increase in problems associated with drug interactions and treatment adherence
− High pressure of day hospital care

**Table 3 cancers-14-03968-t003:** Characteristics of personal protection equipment according to the risk of drug administration.

Administration Route	Gloves	Gown	Respiratory Protection	Ocular Protection
Intramuscular, subcutaneous, intradermal, intrathecal	Yes	Yes ^1^	FFP3	Yes
Intravenous (with closed system)	Yes	No	No	No
Inhalation	Yes	Yes	FFP3	Yes
Topical, vaginal, rectal	Yes (2 pairs)	Yes ^1^	Yes ^2^	No ^3^
Oral (enteral and non-fractioned tablets)	Yes	No	No	No
Oral (tablets, capsules, oral solutions with manipulation)	Yes	Yes ^4^	FFP3	Yes ^3^
Intravesical	Yes	Yes ^1^	FFP3	Yes
Intraperitoneal	Yes (2 pairs)	Yes ^5^	FFP3	Yes
Ophthalmic	Yes	Yes ^1^	FFP3	Yes

^1^ Non-sterile gown resistant to liquids in sleeves and the chest area; ^2^ surgical mask; ^3^ wear glasses if there is sprinkle risk; ^4^ simple gown; ^5^ sterile waterproof gown.

**Table 4 cancers-14-03968-t004:** Distinctive features of bladder cancer in elderly patients.

Bladder Cancer	Elderly Patients with Bladder Cancer
− Tumor of elderly people	− Reduced physiological reserve; more aggressive behavior; different treatment response pattern
− Median age at diagnosis 72 years (range 65–74)	− Main objective: to maintain the patient’s quality of life (even at the expense of renouncing to radical treatment)
− Median age at mortality 79 years (range 75–84)	− Curative treatment is offered less frequently because of the patient’s characteristics, tumor features, and doctor’s beliefs
− Frequent comorbid conditions which make patients ineligible for standard treatment with cisplatin (hearing loss, renal failure, heart disease)	− Age is a poor prognostic factor and is related to a worse response to BCG therapy and higher risk of BCG-associated toxicity (intravascular dissemination and sepsis, respiratory failure, cardiovascular collapse)
− Highly symptomatic neoplasm; toxic treatment (cystectomy, cisplatin chemotherapy)	− Disseminated disease: more than 40% of patients aged ≥70 years have a renal clearance <60 mL/min and associated cardiac and renal dysfunction
− Main challenge: balance between undertreatment and overtreatment maintaining the quality of life	
− One of the highest lifetime expensive malignancies (economic and suffering for the patient)	

## Data Availability

Data supporting recommendations and collected by consensus of the expert members of the Spanish Oncology Genitourinary (SOGUG) Working Group are available upon request to the corresponding author.

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
