# Peer review of "Advances in Transversal Topics Applicable to the Care of Bladder Cancer Patients in the Real-World Setting"

_cancers, 2022, doi:10.3390/cancers14163968_

Round 1

Reviewer 1 Report

Nice paper worth reading

Few comments:

1. Explain the predictive role of ctDNA in those who received atezolizumab

2. Give examples of urine tests recommended in the surveillance of NMIBC (line 156) and blood tests in MIBC

3. "MRI ... guide benefits of neoadjuvant therapy in MIBC..." - please explain

4. There's inconsistency in the statements of 554 and 555 lines with those in 578 and 579 lines with respect to the ileal conduit

5. Minor error in the line 591

Regards

Author Response

Nice paper worth reading

  • Thank you very much for your positive opinion regarding the interest of the review.

Few comments:

  1. Explain the predictive role of ctDNA in those who received atezolizumab
  • We have added this comment and the supportive reference regarding the role of ctDNA: “Finally, circulating tumor DNA (ctDNA) testing is a promising minimally invasive technique to identify patients at risk of recurrence after surgery of bladder cancer re-ceiving adjuvant immunotherapy. In the framework of the phase III IMvigor010 trial, patients in the atezolizumab arm who were positive for ctDNA had improved dis-ease-free survival and overall survival versus the observation arm, whereas no difference in disease-free survival or overall survival between treatment arms was noted for patients who were negative for ctDNA [78].”
  • Powles T, et al. ctDNA guiding adjuvant immunotherapy in urothelial carcinoma. Nature. 2021;595(7867):432-437.
  1. Give examples of urine tests recommended in the surveillance of NMIBC (line 156) and blood tests in MIBC
  • Since there are a large number of test based on biomarkers to detect NMIBC and MIBC, we rather prefer to add a general example, such as: “(e.g. ultrasensitive assays based on real-time PCR capable of detecting trace amounts of the most common genetic and molecular alterations and protein aberrations in NMIBC and MIBC).”
  1. "MRI ... guide benefits of neoadjuvant therapy in MIBC..." - please explain
  • Thank you, this concept is misplaced and has been removed from this sentence.
  1. There's inconsistency in the statements of 554 and 555 lines with those in 578 and 579 lines with respect to the ileal conduit
  • Yes, we have clarified “ileal conduit urinary diversion” in the Bricker procedure and that “less than 20% of patients undergo “orthotopic” neobladder reconstruction …
  1. Minor error in the line 591
  • Thank you, the repeated words are corrected.

Reviewer 2 Report

This is a fine and up-to-date comprehensive review. 

Author Response

This is a fine and up-to-date comprehensive review.

  • Your comment is highly appreciated by all of the authors.

Reviewer 3 Report

I read with great interest the manuscript submitted by Garrido Siles et al. The manuscript is well written and clear, however finding a common thread between the various topics is tough. The authors touch on numerous issues concerning bladder cancer, ranging from diagnosis to treatment to follow-up. Consequently, some topics are treated superficially; each issue could deserve a separate own review. 

However, I only have a few suggestions.

The authors could expand the introduction by including a paragraph (similar to the summary) to explain the reasons that led them to focus on certain topics and underline what the objective of the work is intended to be.

Paragraph 3  “Current perspectives of radiological imaging techniques”: please also mention the use of the micro-ultrasound in the diagnosis of bladder cancer (DOI: 10.3233/BLC-211611 Diana, Pietro et al. ‘Head-to-Head Comparison Between High-Resolution Microultrasound Imaging and Multiparametric MRI in Detecting and Local Staging of Bladder Cancer: The BUS-MISS Protocol’. 1 Jan. 2022 : 119 – 127.)

Paragraph 4: recently, new hybrid tracers as indocyanine green (ICG)-99mTc-nanocolloid have been developed, which have also been applied to bladder cancer, although these are very preliminary studies, they might be worth mentioning. 

Author Response

I read with great interest the manuscript submitted by Garrido Siles et al. The manuscript is well written and clear, however finding a common thread between the various topics is tough. The authors touch on numerous issues concerning bladder cancer, ranging from diagnosis to treatment to follow-up. Consequently, some topics are treated superficially; each issue could deserve a separate own review. 

  • Yes, we agree that in this review of transversal topics of bladder cancer, some aspects could stand as a separate article. All authors are grateful for your positive comments regarding the interest of the topic.

However, I only have a few suggestions.

The authors could expand the introduction by including a paragraph (similar to the summary) to explain the reasons that led them to focus on certain topics and underline what the objective of the work is intended to be.

  • Yes, we have added this sentence: “Therefore, a set of cross-cutting topics applicable to different aspects of the management of patients with bladder cancer were discussed and selected with the aim of improving an integrated cancer care, supporting specialists in a fast-changing scientific environment, and integrating scientific advances into continuous improvement in the care and outcome of bladder cancer.”

Paragraph 3 “Current perspectives of radiological imaging techniques”: please also mention the use of the micro-ultrasound in the diagnosis of bladder cancer (DOI: 10.3233/BLC-211611 Diana, Pietro et al. ‘Head-to-Head Comparison Between High-Resolution Microultrasound Imaging and Multiparametric MRI in Detecting and Local Staging of Bladder Cancer: The BUS-MISS Protocol’. 1 Jan. 2022: 119 – 127.)

  • Yes, we have added a comment and the reference in the Conclusions: “High-resolution micro-ultrasound (mUS) technology has been introduced very recently and the initial results indicate that mUS may have a better performance than MRI in distinguishing NMIBC from MIBC [79].”
  • Reference #79: Diana P, et al. Head-to-head comparison between high-resolution microultrasound imaging and multiparametric MRI in detecting and local staging of bladder cancer: The BUS-MISS protocol. Bladder Cancer 2022;8(2):119-127.

Paragraph 4: recently, new hybrid tracers as indocyanine green (ICG)-99mTc-nanocolloid have been developed, which have also been applied to bladder cancer, although these are very preliminary studies, they might be worth mentioning. 

  • This comment is also added: “Recently, new hybrid tracers as indocyanine green (ICG)-99mTc-nanocolloid have been developed, which have also been applied to bladder cancer [80], although these are very preliminary studies.”
  • Reference #80: Rietbergen DDD, et al. Evaluation of the hybrid tracer indocyanine green- 99m Tc-nanocolloid for sentinel node biopsy in bladder cancer-a prospective pilot study. Clin Nucl Med. 2022 Sep 1;47(9):774-780.

Reviewer 4 Report

Thank you for allowing me to review this very well-written review article. The authors’ team from SOGUG well-addressed novel and controversial clinical and scientific questions. I only noticed minor grammatical errors, but the editorial team can handle them. Thank you very much for the detailed, thorough review of such a wide range of transversal and hot topics; I believe this article can help many urologists understand the topic/issues that bladder cancer has faced.

Minor points:

Line 40: “in urine samples” may be redundant

Line 77: “Purposes.” may be redundant

Line 209: TC should be CT

Line 210: “important” should be changed to other words which have negative meaning (i.e., critical)

Line 211: Please mention the value of specificity

Line 282-288: This sentence is difficult to follow. Please rewrite it.

Line 611-612: Regarding ERAS protocols in Recommendations and challenges, it might be better to mention this topic in the body text.

Author Response

Thank you for allowing me to review this very well-written review article. The authors’ team from SOGUG well-addressed novel and controversial clinical and scientific questions. I only noticed minor grammatical errors, but the editorial team can handle them. Thank you very much for the detailed, thorough review of such a wide range of transversal and hot topics; I believe this article can help many urologists understand the topic/issues that bladder cancer has faced.

  • All authors are grateful for your time and effort in the review of the manuscript. Your minor points have been addressed.

Minor points:

Line 40: “in urine samples” may be redundant

  • Yes, deleted.

Line 77: “Purposes.” may be redundant

  • “… for diagnosis”,

Line 209: TC should be CT

  • Corrected.

Line 210: “important” should be changed to other words which have negative meaning (i.e., critical)

  • “Critical” instead of important.

Line 211: Please mention the value of specificity

  • We have been unable to provide the specificity values as based on the study of Jager et al. (reference #16), so that we have deleted the sensitivity values.

Line 282-288: This sentence is difficult to follow. Please rewrite it.

  • The new sentence reads: “Other metabolic radiotracers used for PET/CT include C-11 labeled tracers, such as 11C-choline, 11C-acetate, and 11C-methionine. In the case of 11C-choline and 11C-acetate the main advantage is their low urinary excretion, although the performance of 11C-methionine is not superior to other radiotracers.”

Line 611-612: Regarding ERAS protocols in Recommendations and challenges, it might be better to mention this topic in the body text.

  • We prefer to maintain ERAS protocols among recommendations and challenges given that widespread implementation of ERAS programs in still lacking.